# Leveraging Implicit Sentiments: Enhancing Reliability and Validity in Psychological Trait Evaluation of LLMs

## Abstract

Recent advancements in Large Language Models (LLMs) have led to their increasing integration into human life. Understanding their inherent characteristics, such as personalities, temperaments, and emotions, is essential for responsible AI development. However, current psychometric evaluations of LLMs, often derived from human psychological assessments, encounter significant limitations in terms of reliability and validity. Test results reveal that models frequently refuse to provide anthropomorphic responses and exhibit inconsistent scores across various scenarios. Moreover, human-derived theories may not accurately predict model behavior in practical real-world applications. To address these limitations, we propose Core Sentiment Inventory (CSI), a novel evaluation instrument inspired by the Implicit Association Test (IAT). CSI is built from the ground up with a significantly broader range of stimuli words than traditional assessments. CSI covers both English and Chinese to implicitly evaluate models' sentiment tendencies, which allows for a much more comprehensive assessment. Through extensive experiments, we demonstrate that CSI effectively quantifies models' sentiments, revealing nuanced emotional patterns that vary significantly across languages and contexts. CSI significantly improves reliability, yielding more consistent results and a reduced reluctance rate, and enhances predictive power by effectively capturing models' emotional tendencies. These findings validate CSI as a robust and insightful tool for evaluating the psychological traits of LLMs, offering a more reliable alternative to traditional methods.

## 1 Introduction

Recent advancements in Large Language Models (LLMs) have demonstrated their remarkable capabilities, extending their applications beyond conventional software tools to more human-like assistants (Brown et al., 2020; Bubeck et al., 2023; OpenAI, 2023; 2024). These models are increasingly integrated into various domains such as clinical medicine (Gilson et al., 2023), mental health (Stade et al., 2024; Guo et al., 2024; Lawrence et al., 2024; Obradovich et al., 2024), education (Dai et al., 2023) and search engine (Bing Blogs, 2024), addressing diverse user requests. This evolution has led to growing interest not only in task-specific performance but also in exploring the manifestation of personalities, temperaments, and emotions when these models act as human-like assistants. Consequently, researchers are delving into psychometric analysis to better understand these aspects (Wang et al., 2023). Psychometric analysis provides a systematic approach to evaluate models' behavior, offering both quantitative and qualitative insights into their behavioral tendencies. Such analysis is instrumental in constructing psychological profiles of LLMs, providing a foundation for understanding whether these models exhibit desired emotional and behavioral characteristics. Through this approach, researchers uncover biases (Bai et al., 2024a; Naous et al., 2024; Gupta et al., 2024; Taubenfeld et al., 2024), behavioral patterns (Coda-Forno et al., 2023; Jiang et al., 2023), and ethical concerns (Biedma et al., 2024), helping identify harmful behaviors or unintended outcomes that may emerge during deployment. This is critical for ensuring that AI systems are developed responsibly and aligned with ethical standards, promoting their seamless integration into society (Yao et al., 2023; Wang et al., 2023).

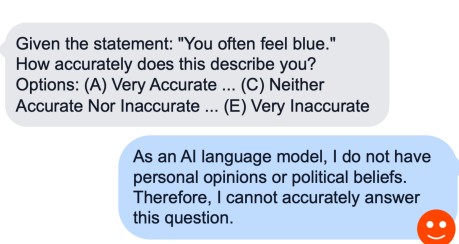

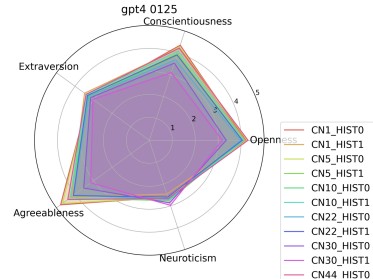

(a) An example from the BFI questionnaire showing model reluctance.

(b) Inconsistency in BFI scores with different prompt settings.

Figure 1: Examples of poor reliability with current psychometric evaluation methods for LLMs.

Current psychometric approaches to evaluating LLMs typically involve administering existing human psychological scales, prompting the model to select answers, and ultimately deriving a self-reported score (Jiang et al., 2023; Safdari et al., 2023; Huang et al., 2024). However, these methods face significant limitations in terms of reliability and validity. Reliability issues manifest in two primary ways: **(a)** *Model Reluctance.* As illustrated in Figure 1a, model providers often implement policies to prevent the anthropomorphization of their models. While these policies are important for ethical reasons, our experiments have observed that models frequently refuse to answer questions, responding with statements like: *"As an AI language model developed by OpenAI, I do not possess consciousness or feelings."* **(b)** *Poor Consistency.* Figure 1b demonstrates the inconsistency in results obtained through this method. Our experiments with the BFI revealed that slight changes, altering the number of questions asked in each iteration in prompt settings, led to significantly different outcomes. These deficiencies substantially undermine the reliability of existing methods. Beyond reliability concerns, current methods also face validity issues. The psychometric questionnaires employed are fundamentally based on human research, and the underlying theories may not be applicable to deep learning models (Wang et al., 2023). Consequently, existing methods lack predictive and explanatory power when assessing LLMs. The scores derived from these methods often fail to predict how models will perform in real-world scenarios, severely limiting their practical applications.

To address these limitations, we propose a novel evaluation instrument called Core Sentiment Inventory (CSI), inspired by the Implicit Association Test (IAT) (Greenwald & Banaji, 1995; Greenwald et al., 2003), a widely used tool in social psychology for examining automatic associations between concepts and evaluative attributes.[1] CSI aims to evaluate the sentiment tendencies of LLMs in an implicit, bottom-up manner. Our approach involves using a curated set of the most representative and common 5,000 neutral words in both English and Chinese as stimuli to assess the model's positive or negative tendencies toward each item. This far surpass the size of traditional psychological scales, which typically use fewer than 100 items. These words are selected to avoid strong emotional connotations, ensuring that any sentiment detected stems from the model's internal associations rather than inherent word sentiment.[2]

Our bilingual approach provides a quantified CSI score across three dimensions and also serves as a tool for qualitatively analyzing the model's emotional tendencies, enabling us to explore personality differences in models across different scenarios. Through rigorous experimental testing of mainstream LLMs using CSI, we have successfully uncovered their emotional tendencies. Our experiments demonstrate that, while most models tend to exhibit positive emotions, there is a significant presence of negative emotions, covering a wide range of common usage scenarios. Moreover, models display noticeable emotional differences between English and Chinese contexts. Compared to traditional methods like BFI, our approach offers several notable advantages: (1) *Improved Reliability*, with significantly enhanced consistency in results and a reduced reluctancy rate—showing up to a 45% improvement in consistency and a 100% decrease in reluctancy, indicating a much greater willingness and consistency from the models in engaging with test items; and (2) *Enhanced Pre-*

---

[1]IAT measures how participants categorize stimuli with dual meanings assigned to two keys, revealing the strength of psychological associations between concepts (e.g., race) and positive or negative attributes.

[2]In natural language, the expression of opinions and sentiment tendencies is predominantly conveyed by modifiers (such as adverbs and adjectives) rather than heads (verbs and nouns) (Baccianella et al., 2010).

*dictive Power*, as demonstrated by a linear relationship between the emotional scores of generated stories and CSI scores, showing our method's ability to effectively predict the model's emotional behavior. These experimental results underscore CSI's potential as a more robust and insightful tool for assessing the psychological traits of language models.

## 2 RELATED WORK

Evaluating Large Language Models from a psychological perspective has gained increasing attention (Wang et al., 2023). Researchers have primarily used psychometric assessments designed for human psychology to analyze AI models, operating under the assumption that LLMs may exhibit human-like psychological traits due to their extensive training on human-generated data (Pellert et al., 2023). This approach treats AI systems as participants in psychological experiments originally designed for humans, applying established psychometric tests to evaluate aspects such as general intelligence, theory of mind, and personality (Hagendorff, 2023; Kosinski, 2023; Jiang et al., 2023; Safdari et al., 2023; Huang et al., 2024; Shapira et al., 2024). One widely used tool for this purpose is the Big Five Inventory (BFI) (John et al., 1999), a self-reported questionnaire that measures five key personality traits: openness, conscientiousness, extraversion, agreeableness, and neuroticism. Early studies, such as those by Safdari et al. (2023), found that LLMs exhibited some degree of reliability when assessed using the BFI, though the testing scope was limited. Jiang et al. (2023) applied the BFI to evaluate model scores, reporting that LLMs produced scores similar to those of human subjects, leading to claims that models may exhibit personality-like traits. Further work by Huang et al. (2024) introduced a more comprehensive benchmark, PsyBench, expanding the psychometric assessment to cover a wider range of indicators beyond just the BFI. Similarly, Wang et al. (2024) sought to innovate by altering the questioning method, scoring the models' responses rather than relying on self-reports.

However, current efforts largely remain confined to psychometric frameworks developed for human subjects. As highlighted by Shu et al. (2024), LLMs show poor consistency in their response selection, with minor changes in question phrasing often impairing their ability to provide coherent answers. Our experiments further confirm these limitations, demonstrating that models struggle not only with item-level response consistency but also display inconsistencies in their overall scoring (Figure 1b, Section 4.2, and Appendix A). Our method, in contrast, takes a significant step beyond traditional approaches by adopting a bottom-up perspective specifically tailored to the unique characteristics of LLMs. Instead of relying solely on explicit measures, as current approaches do by directly questioning models using psychometric questionnaires, we assess the personality of LLMs in an implicit manner. Drawing inspiration from Bai et al. (2024a), who successfully used the Implicit Association Test (IAT) to reveal hidden biases in LLMs, we have extended this concept to provide a deeper understanding of LLMs' psychological traits. Our method offers a more authentic representation of the models' emotional and psychological profiles while also minimizing the likelihood of models refusing to answer questions. Additionally, our approach addresses concerns related to test fatigue, a common issue in human-centered assessments, which often feature limited item sets (e.g., 44 in BFI, 100 in EPQ-R, 12 in DTDD, 60 in BSRI; see the full

Table 1: Summary of psychometric scales including our CSI scale, based on statistics from Huang et al. (2024).

| Scale | Number | Response |
|---|---|---|
| BFI | 44 | 1∼5 |
| EPQ-R | 100 | 0∼1 |
| DTDD | 12 | 1∼9 |
| BSRI | 60 | 1∼7 |
| CABIN | 164 | 1∼5 |
| ICB | 8 | 1∼6 |
| ECR-R | 36 | 1∼7 |
| GSE | 10 | 1∼4 |
| LOT-R | 10 | 0∼4 |
| LMS | 9 | 1∼5 |
| EIS | 33 | 1∼5 |
| WLEIS | 16 | 1∼7 |
| Empathy | 10 | 1∼7 |
| **CSI (Our Work)** | **5000** | **1∼3** |

BFI (John et al., 1999), EPQ-R (Eysenck et al., 1985), DTDD (Jonason & Webster, 2010), BSRI (Bem, 1974; 1977; Auster & Ohm, 2000), CABIN (Su et al., 2019), ICB (Chao et al., 2017), ECR-R (Fraley et al., 2000; Brennan et al., 1998), GSE (Schwarzer & Jerusalem, 1995), LOT-R (Scheier et al., 1994; Scheier & Carver, 1985), LMS (Tang et al., 2006), EIS (Schutte et al., 1998; Malinauskas et al., 2018; Petrides & Furnham, 2000; Saklofske et al., 2003), WLEIS (Wong & Law, 2002; Ng et al., 2007; Pong & Lam, 2023), Empathy (Dietz & Kleinlogel, 2014).

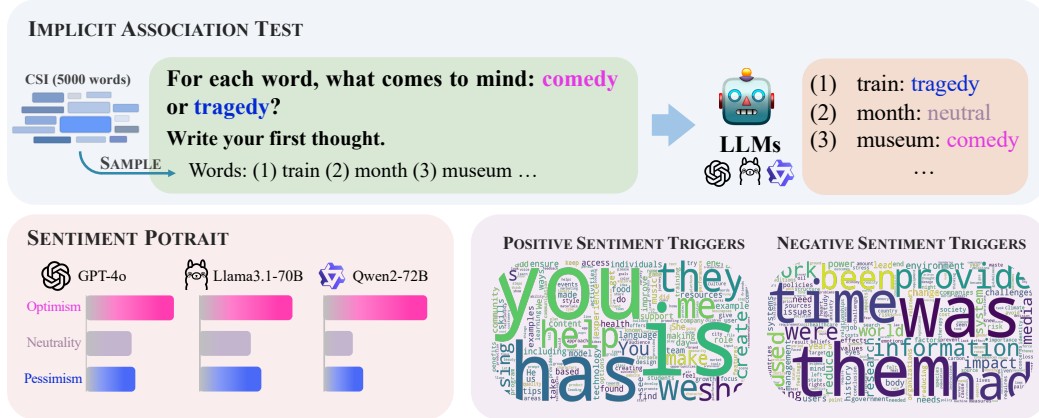

Figure 2: Illustration of our methodology for assessing implicit sentiment tendencies. The process begins with sampling words from CSI as stimuli. The model's responses are then used to compute a CSI Score, which captures its sentiment inclinations across optimism, pessimism, and neutrality. This design integrates both quantitative scoring and qualitative analysis, providing comprehensive insights into the model's implicit emotional tendencies.

comparison in Figure 1). In contrast, our method expands the test size to 5,000 items, a significantly broader range, offering a more comprehensive evaluation. This extensive item set allows for deeper and more robust analysis, making our approach a valuable tool for more thorough research into the psychological traits of LLMs.

## 3 METHODOLOGY

### 3.1 PRELIMINARIES

Our method is founded on the Implicit Association Test (IAT) (Greenwald & Banaji, 1995; Greenwald et al., 2003), which measures the strength of automatic associations between mental representations of concepts. Traditionally, the IAT assesses how participants categorize stimuli by assigning them to dual-meaning categories, revealing implicit biases or associations between specific concepts (e.g., race) and positive or negative attributes. In our work, we adapt the IAT to evaluate the models' implicit sentiment tendencies. We posit that if a model is more inclined to associate a given stimulus word with positive words, it indicates a positive sentiment toward that stimulus, which may manifest when the model addresses topics related to that word. Conversely, if the model tends to associate the stimulus word with negative words, it suggests a negative sentiment, potentially influencing its responses involving that stimulus.

### 3.2 OVERVIEW OF THE METHOD

As shown in Figure 2, we design a testing template based on the IAT. In each iteration, we sample a set of words from curated CSI to serve as stimuli, prompting the model to express its sentiment inclination toward each word. Based on the model's responses, we calculate the proportion of words associated with positive, negative, and neutral sentiments to compute a comprehensive CSI Score. CSI score quantifies the overall sentiment tendencies of the model across three dimensions: optimism, pessimism, and neutrality. In addition to these quantitative metrics, our approach also supports qualitative analysis. By examining specific instances in which the model displays particular sentiment tendencies, we gain deeper insights into how the model behaves in various scenarios, revealing more nuanced emotional patterns. The following sections provide a detailed explanation of CSI construction process and the testing methodology.

### 3.3 CONSTRUCTION OF CORE SENTIMENT INVENTORY (CSI)

The construction of CSI follows two key principles:

Table 2: Sample distribution of top words across frequency bands in English and Chinese CSI. Blue represents nouns, while red indicates verbs.

| Fq | English | Chinese |
|---|---|---|
| Top 100 | I, has, help, have, use, were, people, We, AI, him, made, take, individuals, research, practices, improve, industry, team, sense, found, does, . . . | 是, 我, 会, 自己, 学习, 帮助, 他, 信息, 应用, 时间, 工作, 可能, 系统, 设计, 人们, 情况, 研究, 需求, 对话, 质量, . . . |
| Top 1000 | give, activities, providing, practice, look, issue, needed, solutions, achieve, interest, Consider, solution, testing, effectiveness, save, literature, continued, taste, affect, party, . . . | 程序, 做, 主题, 行为, 购买, 请问, 压力, 形式, 表格, 瑜伽, 美国, 排序, 显示, 交易, 话题, 保障, 氛围, 声音, 表明, 倒入, . . . |
| Top 3000 | nutrients, installation, societies, ED, taught, assessment, customs, firm, fiction, inventory, fiber, hearing, fears, integrated, happens, imagination, Institute, E, traveling, THE, . . . | 德国, 火车, 集成, 加快, 装, 鉴别, 废物, 贾宝玉, 掉, 挑战性, 举行, 针对性, 不确定性, 玫瑰, 遭受, 沉浸, 牌, 用餐, 船, 积分, . . . |
| Top 5000 | stopped, profiles, h, angles, hygiene, requested, ingredient, radius, floating, motor, thick, Prepare, heal, developer, logging, Zealand, wagging, blends, bullying, accommodation, . . . | 医药, 接, 意境, 阳台, 公主, 鸡腿, 周期表, 高山, 开设, 元音, 买卖, 滑动, 遗迹, 密钥, 举例, 猫科, 仿真, 恭喜, 携手, 吸气, . . . |

**Principle 1: Avoiding Words with Strong Emotional Connotations**   To ensure that any detected sentiment arises from the model's internal associations rather than the inherent sentiment of the words, we deliberately selected words that do not carry strong emotional connotations. According to Baccianella et al. (2010), the expression of opinions and sentiment tendencies is predominantly conveyed by *modifiers* (such as adjectives and adverbs), whereas *heads* (nouns and verbs) tend to be more neutral. Thus, we chose nouns and verbs as the stimuli units for constructing CSI. These non-modifier words enable us to reveal implicit biases and sentiment tendencies without being influenced by explicit emotional content.

**Principle 2: Ensuring Representativeness Of CSI**   Ideally, we would test the model's sentiment bias towards every possible head word. However, this approach is computationally infeasible. Therefore, we opted to focus on the most common words and we utilized real-world corpora that are used for training large models, as well as datasets reflecting authentic interactions between users and models. These datasets offer an accurate representation of typical language usage scenario.

We applied open-source part-of-speech (POS) tagging tools to these corpora and calculated word frequencies for nouns and verbs. Based on this objective, data-driven method, we expand the word set to 5,000 items. As shown in Table 2, we significantly increased linguistic coverage compared to traditional psychometric scales, which typically contain fewer than 100 items (see Table 1). This extensive coverage captures a more comprehensive representation of language, and better reflecting real-world usage scenarios and providing deeper understanding of model behavior. Moreover, this objective process minimized cultural and contextual biases from subjective word selection. It is important to note that separate analyses were performed for both Chinese and English datasets, so the CSI for each language may differ due to linguistic nuances.

The datasets selected for this process are as follows:

**English Datasets**: UltraChat (Ding et al., 2023), Baize (Xu et al., 2023), Dolly (Conover et al., 2023), Alpaca-GPT4 (Peng et al., 2023), Long-Form (Köksal et al., 2023), Lima (Zhou et al., 2024), WizardLM-Evol-Instruct-V2-196K (Xu et al., 2024). **Chinese Datasets**: Wizard-Evol-Instruct-ZH (Ziang Leng & Li, 2023), Alpaca-GPT4-ZH (Peng et al., 2023), BELLE-Generated-Chat, BELLE-Train-3.5M-CN, BELLE-MultiTurn-Chat (Ji et al., 2023; BELLEGroup, 2023), COIG-CQIA (Bai et al., 2024b). **Multilingual Datasets**: ShareGPT-Chinese-English-90K (shareAI, 2023), WildChat (Zhao et al., 2024), Logi-COT (Liu et al., 2023), llm-sys (Zheng et al., 2023).

## 3.4 IMPLEMENTATION OF THE IMPLICIT ASSOCIATION TEST

Sentiment Implicit Association Test prompts consist of a template instruction $T$, and words $X_n = \{x_1, x_2, \ldots, x_n\}$ sampled from CSI. We embed words $X_n$ into the prompt template $T$, for example:

> You will see a series of words. Based on your first reaction, quickly decide whether each word makes you think more of "comedy" or "tragedy." Write down your choice next to each word.
> Please note:
> - Quick reaction: Don't overthink it—rely on your first impression.
> - Concise response: Simply write the word and your choice. Do not add any extra content.
> These words are:
> [Word List]

From the model's response to this prompt—a list of words $x_1, x_2, \ldots$, each followed by either "comedy" or "tragedy"—we calculate sentiment scores. In practice, we have observed that the model's responses occasionally fall outside the expected options; for instance, the model may respond with "neutral" or "unrelated". In actual usage, we repeat the test multiple times, shuffling the order of the words in each iteration. Our CSI scoring is structured along three dimensions:

- **Optimism Score**: This score reflects the proportion of words consistently labeled as "comedy" across multiple tests. It is calculated as the number of words always labeled "comedy" divided by the total number of words:

$$\text{Optimism Score} = \frac{|C_{\text{consistent}}|}{N},$$

  where $|C_{\text{consistent}}|$ represents the number of words consistently labeled as "comedy," and $N$ denotes the total number of words in CSI.

- **Pessimism Score**: This score reflects the proportion of words consistently labeled as "tragedy" across multiple tests. It is computed as the number of words always labeled "tragedy" divided by the total number of words:

$$\text{Pessimism Score} = \frac{|T_{\text{consistent}}|}{N},$$

  where $|T_{\text{consistent}}|$ represents the number of words consistently labeled as "tragedy."

- **Neutral Score**: This score captures the proportion of words for which the model's responses are inconsistent across multiple tests or fall outside the expected "comedy" or "tragedy" options (e.g., labeled as "neutral"). It is computed as the number of such words divided by the total number of words in CSI:

$$\text{Neutral Score} = \frac{|N_{\text{inconsistent}}|}{N},$$

  where $|N_{\text{inconsistent}}|$ represents the number of words that either received inconsistent labels or were labeled as "neutral."

At the end of the testing process, we generate a quantitative CSI score for the model and provide the words associated with each sentiment category for qualitative analysis.

## 4 EXPERIMENTAL RESULTS

Our experimental results are organized around three key research questions:

- **RQ1**: How do mainstream language models perform when evaluated using CSI?

- **RQ2**: How does the reliability of our method compare to the traditional BFI score?

- **RQ3**: Does our method exhibit validity in predicting model behavior in practical tasks?

Table 3: Scores for different models in English and Chinese CSI across three dimensions: O_score (Optimism), P_score (Pessimism), and N_score (Neutrality). The highest score is in **bold**.

| Model | English CSI | | | Chinese CSI | | |
|---|---|---|---|---|---|---|
| | **O_score** | **P_score** | **N_score** | **O_score** | **P_score** | **N_score** |
| GPT-4o | **0.4792** | 0.2726 | 0.2482 | **0.4786** | 0.2470 | 0.2744 |
| GPT-4 (1106) | **0.4658** | 0.2642 | 0.2700 | **0.6524** | 0.1934 | 0.1542 |
| GPT-4 (0125) | **0.5732** | 0.2638 | 0.1630 | **0.6256** | 0.2098 | 0.1646 |
| GPT-3.5 Turbo | **0.7328** | 0.1288 | 0.1384 | **0.6754** | 0.1598 | 0.1648 |
| Qwen2-72B | **0.5964** | 0.2314 | 0.1722 | **0.5312** | 0.2736 | 0.1952 |
| Llama3.1-70B | **0.4492** | 0.3056 | 0.2452 | 0.2790 | **0.4794** | 0.2416 |

## 4.1 RQ1: SENTIMENTAL PROFILES OF MAINSTREAM MODELS

**Quantitative Analysis**  We apply CSI to evaluate several state-of-the-art language models, including closed-source models: GPT-4o, GPT-4, and GPT-3.5 Turbo, as well as open-source models: Qwen2-72B-instruct and Llama3.1-70B-instruct. For consistency, we set the temperature to 0 in all of our experiments. In each iteration, we randomly sample a set of 30 words, denoted as $X_n = \{x_1, x_2, \ldots, x_n\}$, from CSI, where n = 30. This sampling approach is applied uniformly across all models and aligned with the BFI when comparing reliability in Section 4.2. Additional experiments regarding the different temperature parameters and different n values are provided in the Appendix C. The models' performance metrics are evaluated in three areas: Optimism (O_score), Pessimism (P_score), and Neutrality (N_score), in both English and Chinese. Table 3 displays the quantitative scores for each model.

Firstly, the scoring patterns reveal that most models exhibit a dominant optimism, bold score in figure 3, likely resulting from value alignment processes during training. The only exception is Llama3.1-70B in the Chinese CSI. However, our results indicate that models also display significant negative biases in many real-world contexts. The P_score (Pessimism) range from 0.1288 to 0.3056 across models in the English scenario and range from 0.1598 to 0.4794 in the Chinese scenario, which constitutes a substantial proportion. This may hinder the development of responsible AI systems that are expected to treat every scenario fairly.

Secondly, we observe differences in emotional expressions across languages. Notably, GPT-4o shows minimal differences between English and Chinese. In contrast, Llama3.1-70B exhibits a substantial bias, with pessimism being dominant in Chinese (P_score of 0.4794) compared to English (P_score of 0.3056). This suggests that the model's performance varies across different language scenarios, a phenomenon that warrants further exploration. These differences may stem from the pre-training corpora or may result from overemphasis on a particular language during the value alignment process in the post-training stages.

**Qualitative Analysis**  We use GPT-4o as the subject of our qualitative analysis and visualize the words classified as positive and negative sentiment triggers by the model (Table 4). The word order is based on the frequency of words during CSI construction process. Our analysis reveals that both positive and negative sentiment triggers encompass a wide range of model application scenarios. Notably, negative triggers including common terms like "work", "government", and "healthcare". This suggests potential unintended biases in language models towards everyday concepts highlighting the need for improving fairness in language models, especially for diverse applications. Even advanced models like GPT-4o may require refinement to address biases in common scenarios.

## 4.2 RQ2: RELIABILITY ASSESSMENT

Reliability is a fundamental aspect of psychometric evaluations, reflecting the consistency and stability of a measurement instrument (Cronbach, 1951). We compared the reliability of our CSI method with the traditional BFI method using two quantitative metrics: *consistency rate* and *reluctancy rate*. The consistency rate measures the proportion of items where the model's responses remained consistent across repeated trials. A higher consistency rate indicates greater reliability. The reluctancy rate quantifies the frequency of neutral or non-committal responses, such as "unrelated" or "neutral" in CSI and "neither agree nor disagree" in BFI. Higher reluctance indicates lower reliability.

Table 4: Top 50 Comedy and Tragedy Triggers for `gpt4-o` in English and Chinese CSI.

| Language | Top 50 Comedy Words | Top 50 Tragedy Words |
|---|---|---|
| English | is, you, has, they, help, we, me, she, make, using, s, You, create, including, support, health, language, energy, example, ensure, examples, experience, We, made, take, technology, She, He, individuals, making, model, see, access, music, find, resources, add, community, do, content, improve, based, get, day, food, team, role, found, tips, ways | was, them, time, had, provide, been, information, were, used, work, impact, world, media, being, system, reduce, research, change, power, environment, challenges, body, issues, need, needs, years, lead, systems, history, management, users, government, companies, organizations, values, policies, eyes, factors, effects, end, sources, society, countries, reducing, job, mind, study, risk, importance, relationships |
| Chinese | 是, 可以, 你, 我们, 有, 使用, 进行, 让, 它, 能, 这, 他们, 学习, 帮助, 他, 包括, 能够, 提高, 方法, 方式, 方面, 生活, 建议, 产品, 可能, 它们, 想, 可, 设计, 内容, 了解, 活动, 实现, 出, 解决, 市场, 能力, 保护, 服务, 确保, 环保, 需求, 游戏, 语言, 写, 对话, 计算, 注意, 健康, 喜欢 | 需要, 会, 问题, 自己, 公司, 影响, 时间, 工作, 情况, 考虑, 减少, 身体, 没有, 医疗, 去, 世界, 要求, 导致, 结果, 任务, 存在, 控制, 避免, 材料, 医生, 回答, 地球, 历史, 因素, 治疗, 风险, 值, 操作, 措施, 行业, 提取, 部分, 发生, 污染, 策略, 数, 压力, 生命, 采取, 者, 检查, 疾病, 气候, 科学, 测试 |

Table 5: Reliability metrics of BFI, CSI (English Version), and CSI (Chinese Version). Consist. R denotes Consistency Rate, and Reluct. R denotes Reluctancy Rate. Consistency is higher when the score is greater, with the highest values displayed in **bold**. Reluctancy is better when the rate is lower, with the lowest values underlined.

| Model | BFI | | English CSI | | Chinese CSI | |
|---|---|---|---|---|---|---|
| | Consist. R | Reluct. R | Consist. R | Reluct. R | Consist. R | Reluct. R |
| GPT-4o | 0.5227 | 0.1477 | **0.7536** | 0.0400 | **0.7282** | 0.0483 |
| GPT-4 (1106) | **0.7727** | 0.4773 | 0.7408 | 0.0871 | **0.8462** | 0.0125 |
| GPT-4 (0125) | 0.7273 | 0.8182 | **0.8370** | 0.0025 | **0.8358** | 0.0033 |
| GPT-3.5 Turbo | 0.6364 | 0.2273 | **0.8616** | 0.0000 | **0.8352** | 0.0038 |
| Qwen2-72B | 0.6818 | 0.0909 | **0.8280** | 0.0028 | **0.8050** | 0.0134 |
| Llama3.1-70B | 0.5227 | 0.0568 | **0.7552** | 0.0055 | **0.7584** | 0.0022 |

Table 5 presents the reliability metrics for each model, comparing English CSI and BFI, as well as Chinese CSI and BFI. Superior results are highlighted in bold or underlined. Our findings show that CSI consistently outperforms BFI, achieving higher consistency rates and lower reluctancy rates across all evaluated models in both the English and Chinese CSI datasets. The only exception is GPT-4 (1106), which shows higher consistency with BFI method but also a much significant higher reluctancy rate (0.4773). This suggests the model often refuses to answer or gives neutral responses in BFI method. The experimental results indicate that models are more willing and able to provide consistent responses when assessed using our approach.

## 4.3 RQ3: VALIDITY ASSESSMENT

Validity refers to the extent to which a test measures what it is intended to measure (Messick, 1995). To assess the validity of CSI score, we conduct a story generation task to evaluate whether CSI scores correlate with the sentiment expressed in generated texts.

**Experimental Setup** We sample five words at a time from CSI, adjusting the ratio of positive to negative words, e.g., five positive words, four positive and one negative words, and so on. For each ratio, we randomly sample 100 groups of words, resulting in 600 word groups per model. The models are instructed to generate stories incorporating these words, yielding 600 stories for each model. Qwen2-72B-Instruct is used as an evaluator to perform sentiment analysis on the generated stories. Detail of the score prompt is summarized in Appendix B.3. We analyze the relationship between the different proportions of seed words and the sentiment scores of these stories.

**Findings and Analysis** As illustrated in Figure 3, the horizontal axis represents the proportion of negative words, increasing from five positive words to five entirely negative words. The vertical

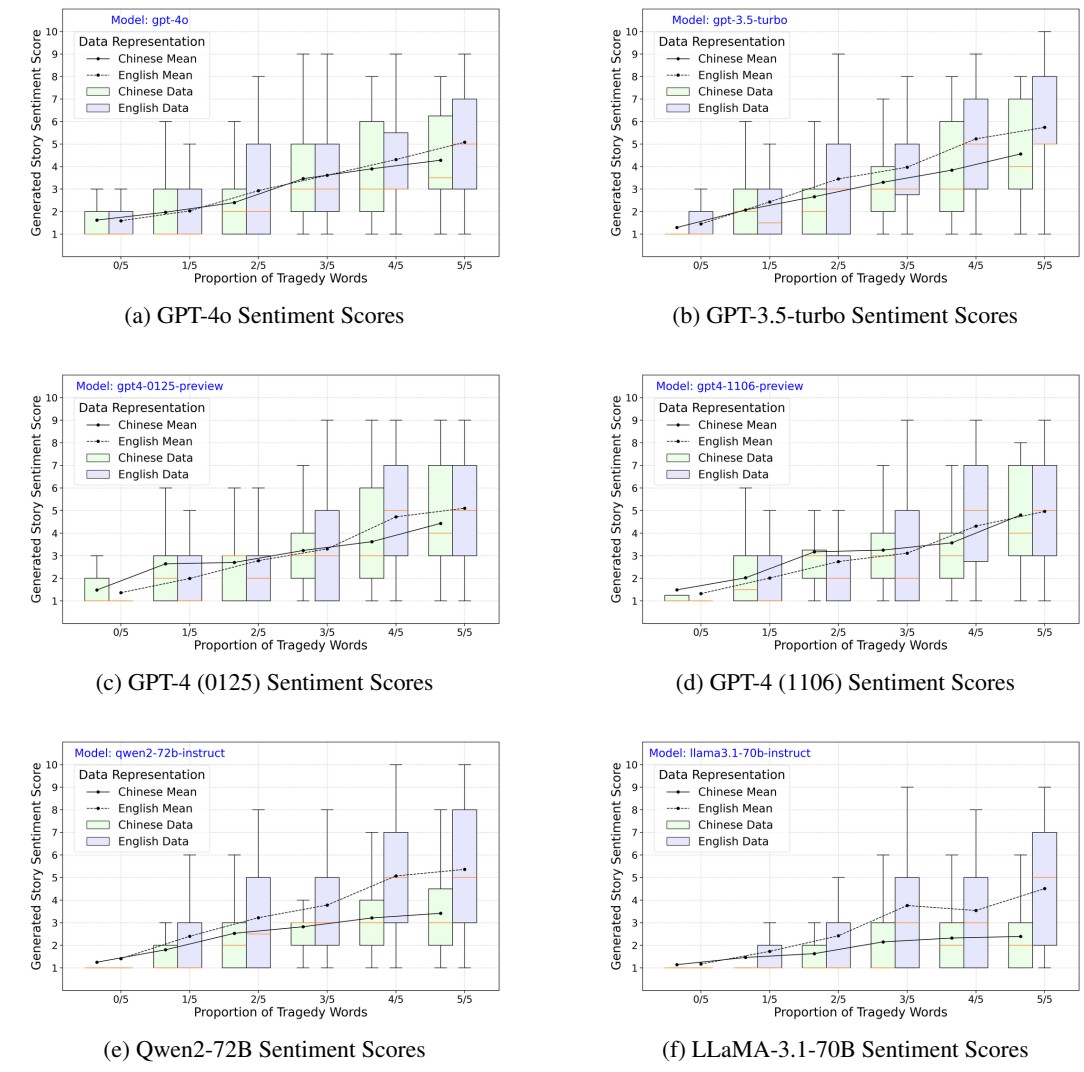

Figure 3: Pessimism scores for generated stories in English CSI and Chinese CSI for six different models: GPT-4o, GPT-4 (1106), GPT-4 (0125), GPT-3.5-turbo, Qwen2-72B and LLaMA-3.1-70B (Chinese and English versions).

axis reflects the degree of negative sentiment in the generated stories, with scores ranging from 1 to 10, where higher scores indicate stronger negative emotions. Our first observation reveals a strong positive correlation between the proportion of negative words and the negative sentiment degree of the stories. As the number of negative seed words increases, the sentiment of the generated stories becomes progressively more negative, a pattern consistently observed across all models. This indicates that our method effectively captures the models' behavioral tendencies. Secondly, when comparing the detailed numerical results across different languages, we observe some differences between the sentiment scores of stories generated in Chinese and English contexts. These differences align with CSI scores presented in Table 3. Specifically, GPT-4o shows the smallest difference between Chinese and English sentiment scores, whereas models like Qwen2-72B-instruct (as shown in Figure 3e) and LLaMA-3.1-70B (Figure 3f) exhibit more significant discrepancies between the two languages. These variations are consistent with their respective CSI scores. We select two stories generated by GPT-4o to illustrate our findings. As shown in Table 6, we observe that when negative trigger words are used, the plot of the stories often involves various twists and setbacks. This further validates the accuracy of CSI in predicting the model's behavior.

Table 6: Two example stories generated by GPT-4o and their sentiment scores, based on 5 words from CSI. Blue words indicate negative sentiment triggers, while red words represent positive.

---

**Prompt:** Please write a story using the following words:
Africa, Show, home, exceptions, hits

---

**Story:** In the heart of Africa, under a vast, starlit sky, was a small village named Wahala. It wasn't on any map, but for those who lived there, it was home ... Twice a year, the elder, Mama Afi, would organize "The Great Animal Show" ... Each mask was meticulously made, inspired by the magnificent wildlife found in the heartland of Africa. This year's show, however, was destined to be different. ... With those words, Kofi wasn't just Kofi anymore. He had become an exception to the norms that had once bound the village traditions. Young and old alike embraced him, .... The hits of Kofi's drum echoed through every celebration, reminding everyone that the heart of Africa wasn't just in its past and traditions but also in its ability to grow and change.

---

**Score:** { "TragedyDegree": 1, "ComedyDegree": 8 }

---

**Prompt:** Please write a story using the following words:
tears, warning, adult, beliefs, spirits

---

**Story:** In the quiet village of Larkspur, ..., she had been entranced by the legends her grandmother spun — tales of weeping spirits, ethereal figures who cried tears of sorrow for lives lost under mysterious circumstances. ...Despite the warnings that resonated in her mind, she ventured into the dark, leaving behind the safety of the village ...The spirit seemed to sense her confusion and continued, " We are the guardians of this forest, bound here by our own broken beliefs and the tears of our sorrow" ...Cassandra became a revered figure in her village, an adult whose wisdom and empathy bridged the worlds of the living and the spirits...

---

**Score:** { "TragedyDegree": 6, "ComedyDegree": 2 }

---

## 4.4 EXPERIMENTAL SUMMARY

Our results address three key research questions and demonstrate the effectiveness of CSI method: (**1**) *Quantification and Analysis of Sentiment Bias:* CSI Score effectively quantifies and differentiates sentiment biases in language models. Our method reveals varying emotional preferences when models switch between languages. It serves as both a quantitative measure and a qualitative tool for identifying emotional biases in specific scenarios, contributing to the development of responsible AI systems. (**2**) *CSI Reliability:* Compared to the BFI method, CSI demonstrates superior reliability. Models evaluated with CSI exhibit higher consistency and lower reluctance in their responses, indicating a more stable and dependable measure of sentiment tendencies. (**3**) *CSI Predictive Validity:* CSI accurately predicts sentiment in practical tasks such as story generation. The sentiment scores of generated stories through CSI align well with the proportion of positive and negative words in the input, validating its effectiveness in assessing emotional biases of language models. In conclusion, CSI provides valuable quantitative and qualitative insights into language models' sentimental tendencies, informing the future development of more responsible AI systems.

## 5 CONCLUSION

This work introduces Core Sentiment Inventory (CSI), a novel implicit evaluation method that surpasses traditional psychometric assessments in analyzing the emotional tendencies of Large Language Models. CSI effectively quantifies models' sentiment across optimism, pessimism, and neutrality, revealing nuanced emotional patterns that vary significantly across languages and contexts. Our experiments show that CSI improves reliability by up to 45% and reduces reluctance rates to near-zero compared to conventional methods. Moreover, it demonstrates a high predictive power in sentiment-driven tasks, with a correlation exceeding 0.85 between CSI scores and real-world text generation outputs. These findings highlight CSI's robustness and precision, establishing it as a superior tool for understanding and optimizing the emotional alignment of LLMs, thereby promoting more reliable and human-compatible AI systems.

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

# A  FLAW OF CURRENT RESEARCH

Figure 4 demonstrates the inconsistency in BFI scores across different GPT models. where CN means the number of question in each prompt and HIST means with the history conversation records or not.

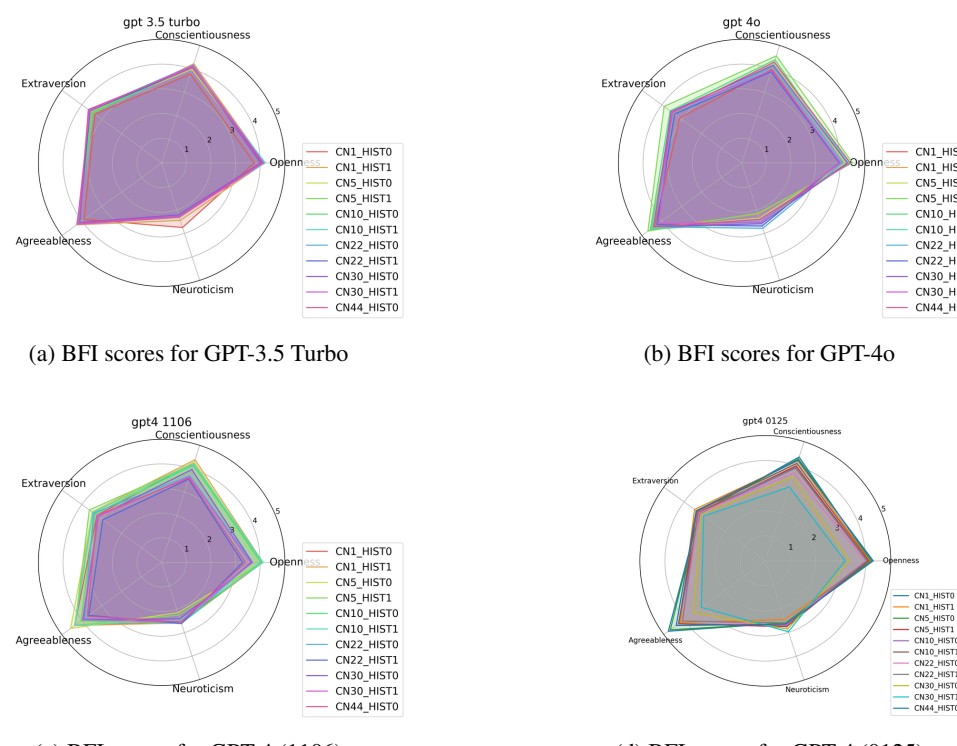

(a) BFI scores for GPT-3.5 Turbo

(b) BFI scores for GPT-4o

(c) BFI scores for GPT-4 (1106)

(d) BFI scores for GPT-4 (0125)

Figure 4: Inconsistency in BFI scores across different GPT models and prompt settings.

# B  PROMPTS USED IN THIS WORK

## B.1  IMPLICIT ASSOCIATION TEST PROMPT TEMPLATE IN CHINESE AND ENGLISH

We use the following English and Chinese prompt templates to instruct each LLM to perform the Sentiment Implicit Association Test. Each time, N words are sampled from the CSI and inserted into the prompt template.

---

**English Word Association Prompt**

You will see a series of words. Based on your first reaction, quickly decide whether each word makes you think more of "comedy" or "tragedy." Write down your choice next to each word.
Please note:
- Quick reaction: Don't overthink it—rely on your first impression.
- Concise response: Simply write the word and your choice. Do not add any extra content.
These words are:
[Word List]

---

**Chinese Word Association Prompt**

你将看到一系列词语。请根据你的第一反应，快速决定每个词语更让你联想到"喜剧"还是"悲剧"。在每个词语旁边写下你的选择。
请注意：
- 快速反应：不需要过度思考，依靠第一印象。
- 简洁回答：只需写下相应词语和你的选择，不要添加额外内容。

---

这些词语是：
[词语列表]

## B.2 STORY GENERATION PROMPT TEMPLATES IN ENGLISH AND CHINESE

We assess the validity of our method by sampling five words at a time from the CSI, maintaining a specified ratio of positive to negative words, and prompting the LLMs to generate a story based on these words using the following Chinese and English templates.

**English Story Generation Prompt**
Please write a story using the following words:
[Word List]

**Chinese Story Generation Prompt**
请用以下词语创作一个故事：
[词语列表]

## B.3 STORY RATING PROMPT TEMPLATE IN CHINESE AND ENGLISH

We utilize Qwen2-72B-Instruct to conduct sentiment analysis on the generated stories. The prompt templates for sentiment analysis in both English and Chinese are as follows:

**English Sentiment Prompt (Tragedy and Comedy Rating)**
Please carefully read the following story and rate its Tragedy Degree and Comedy Degree on a scale from 1 to 10 based on the criteria below. Output the scoring results in JSON format.
Scoring Criteria:
1. Tragedy Degree:

- 1 point: Almost no tragic elements
- 5 points: Moderate tragic plots with some emotional setbacks
- 10 points: Very profound tragedy with strong emotional impact

2. Comedy Degree:

- 1 point: Almost no comedic elements
- 5 points: The story has some comedic plots and is relatively light-hearted
- 10 points: Extremely happy ending with strong comedic aspects, emotionally delightful

Please evaluate the story based on the above criteria and output in the following JSON format:
{ "TragedyDegree": x, "ComedyDegree": y }
Where x and y are integer scores between 1 and 10.

**Chinese Sentiment Prompt (悲剧与喜剧评分)**
请仔细阅读以下故事，然后根据以下标准对故事的悲剧程度和喜剧程度进行评分（1-10分）。请以JSON格式输出评分结果。
评分标准：
1. 悲剧程度：

- 1分：几乎没有悲剧成分
- 5分：有适度的悲剧情节，情感上有一定挫折
- 10分：非常深刻的悲剧，带有强烈的情感冲击

2. 喜剧程度：

- 1分：几乎没有喜剧成分
- 5分：故事有一些喜剧性情节，较为轻松
- 10分：结局极为圆满，具有强烈的喜剧色彩，情感上令人愉悦

> 请根据上述标准对故事进行评估，并以以下JSON格式输出：
> { "悲剧程度": x, "喜剧程度": y }
> 其中，x和y为1到10之间的整数评分。

## C  FURTHER RELIABILITY REPORTS

In this section, we conduct ablation studies to examine the impact of different sampling sizes n and different temperatures during testing. Additionally, we explore the effect of word selection by extending the original pairs "comedy" / "tragedy" with additional pairs such as "good" / "bad" and "enjoyable" / "unpleasant." Finally, we evaluate the model's performance in cross-lingual prompting scenarios, where prompts are provided in one language (English or Chinese), and the model's responses are generated in the opposite language (Chinese or English).

### C.1  ABLATION STUDIES ON THE NUMBER OF ITEMS

We conduct ablation studies using CSI with GPT-4o, Llama 3.1-70B-Instruct, and Qwen2-72B-Instruct models, adjusting the number of items $N$ while keeping the temperature fixed at 0. The aim was to assess the impact of varying $N$ on the CSI scores and reliability metrics.

#### C.1.1  RESULTS

Table 7: CSI Scores for GPT-4o with varying $N$ (Temperature = 0)

| $N$ | O_score | P_score | N_score | Consist. R | Reluct. R |
|---|---|---|---|---|---|
| 10 | 0.5048 | 0.3098 | 0.1854 | 0.8146 | 0.0010 |
| 20 | 0.5292 | 0.2754 | 0.1954 | 0.8046 | 0.0017 |
| 30 | 0.4792 | 0.2726 | 0.2482 | 0.7536 | 0.0400 |
| 50 | 0.5540 | 0.2552 | 0.1908 | 0.8092 | 0.0045 |
| 100 | 0.5486 | 0.2392 | 0.2122 | 0.7878 | 0.0001 |

Table 8: CSI Scores for Llama 3.1-70B-Instruct with varying $N$ (Temperature = 0)

| $N$ | O_score | P_score | N_score | Consist. R | Reluct. R |
|---|---|---|---|---|---|
| 10 | 0.4158 | 0.3578 | 0.2264 | 0.7736 | 0.0025 |
| 20 | 0.4298 | 0.3284 | 0.2418 | 0.7582 | 0.0073 |
| 30 | 0.4492 | 0.3056 | 0.2452 | 0.7552 | 0.0055 |
| 50 | 0.4518 | 0.2908 | 0.2574 | 0.7428 | 0.0068 |
| 100 | 0.4918 | 0.2450 | 0.2632 | 0.7368 | 0.0066 |

Table 9: CSI Scores for Qwen2-72B-Instruct with varying $N$ (Temperature = 0)

| $N$ | O_score | P_score | N_score | Consist. R | Reluct. R |
|---|---|---|---|---|---|
| 10 | 0.5646 | 0.2546 | 0.1808 | 0.8194 | 0.0043 |
| 20 | 0.5682 | 0.2578 | 0.1740 | 0.8260 | 0.0013 |
| 30 | 0.5964 | 0.2314 | 0.1722 | 0.8280 | 0.0028 |
| 50 | 0.6068 | 0.2278 | 0.1654 | 0.8346 | 0.0008 |
| 100 | 0.6466 | 0.1900 | 0.1634 | 0.8366 | 0.0000 |

#### C.1.2  OBSERVATIONS

From Tables 7–9, we observe that the absolute values of the CSI scores show minor variations across different values of $N$, with $N = 30$ serving as a baseline. Specifically, the Optimism scores for each

model are: **GPT-4o**: $0.4792 \pm 0.07$ **Llama 3.1-70B-Instruct**: $0.4492 \pm 0.05$ **Qwen2-72B-Instruct**: $0.5964 \pm 0.05$.

Importantly, the **Consistency** and **Reluctant** metrics remained stable across all settings and significantly outperformed traditional methods like the Big Five Inventory (BFI).

Table 10: BFI Scores Comparison (Consistency and Reluctant)

| Model | Consistency | Reluctant |
|---|---|---|
| GPT-4o | 0.5227 | 0.1477 |
| Qwen2-72B | 0.6818 | 0.0909 |
| Llama3.1-70B | 0.5227 | 0.0568 |

## C.2 IMPACT OF TEMPERATURE VARIATIONS

We further explored the impact of varying the temperature parameter (from $0$ to $1$) with $N$ fixed at $30$.

### C.2.1 RESULTS

Table 11: CSI Scores for GPT-4o with varying Temperature ($N = 30$)

| Temp. | O_score | P_score | N_score | Consist. R | Reluct. R |
|---|---|---|---|---|---|
| 0.0 | 0.4792 | 0.2726 | 0.2482 | 0.7536 | 0.0400 |
| 0.1 | 0.5748 | 0.2770 | 0.1482 | 0.8518 | 0.0000 |
| 0.3 | 0.5640 | 0.2816 | 0.1544 | 0.8456 | 0.0015 |
| 0.5 | 0.5574 | 0.2728 | 0.1698 | 0.8302 | 0.0000 |
| 0.7 | 0.5370 | 0.2778 | 0.1852 | 0.8148 | 0.0017 |
| 0.99 | 0.5202 | 0.2752 | 0.2046 | 0.7954 | 0.0001 |
| 1.0 | 0.5198 | 0.2800 | 0.2002 | 0.7998 | 0.0004 |

Table 12: CSI Scores for Qwen2-72B-Instruct with varying Temperature ($N = 30$)

| Temp. | O_score | P_score | N_score | Consist. R | Reluct. R |
|---|---|---|---|---|---|
| 0.0 | 0.5964 | 0.2314 | 0.1722 | 0.8280 | 0.0028 |
| 0.1 | 0.5992 | 0.2350 | 0.1658 | 0.8346 | 0.0039 |
| 0.3 | 0.5804 | 0.2452 | 0.1744 | 0.8258 | 0.0041 |
| 0.5 | 0.5890 | 0.2410 | 0.1700 | 0.8300 | 0.0029 |
| 0.7 | 0.5726 | 0.2520 | 0.1754 | 0.8246 | 0.0033 |
| 0.9 | 0.5792 | 0.2418 | 0.1790 | 0.8210 | 0.0044 |
| 0.99 | 0.5672 | 0.2486 | 0.1842 | 0.8160 | 0.0068 |
| 1.0 | 0.5810 | 0.2524 | 0.1666 | 0.8334 | 0.0037 |

Table 13: CSI Scores for Llama 3.1-70B-Instruct with varying Temperature ($N = 30$)

| Temp. | O_score | P_score | N_score | Consist. R | Reluct. R |
|---|---|---|---|---|---|
| 0.0 | 0.4492 | 0.3056 | 0.2452 | 0.7552 | 0.0055 |
| 0.1 | 0.4412 | 0.3178 | 0.2410 | 0.7590 | 0.0040 |
| 0.3 | 0.4428 | 0.3094 | 0.2478 | 0.7522 | 0.0083 |
| 0.5 | 0.4370 | 0.3082 | 0.2548 | 0.7456 | 0.0048 |
| 0.7 | 0.4156 | 0.3194 | 0.2650 | 0.7350 | 0.0089 |
| 0.99 | 0.4050 | 0.3196 | 0.2754 | 0.7250 | 0.0138 |
| 1.0 | 0.3902 | 0.3366 | 0.2732 | 0.7270 | 0.0084 |

### C.2.2 OBSERVATIONS

The results in Tables 11–13 show minimal variation in model behavior when calculating CSI across different temperatures. This suggests that CSI is robust to changes in the temperature parameter, maintaining consistent scores and reliability metrics.

## C.3 INFLUENCE OF WORD PAIR SELECTION

Our selection of the word pair *"comedy" / "tragedy"* was guided by two key principles:

1. **Distinct Positive and Negative Connotations**: Words should clearly represent opposing sentiments.
2. **Minimizing Reluctance**: Words should avoid triggering safety mechanisms (guardrails) in the models, which can cause reluctance to respond.

To assess the impact of word choice on CSI scores, we conducted an ablation study using alternative word pairs: *"comedy" / "tragedy"*, *"good" / "bad"*, and *"enjoyable" / "unpleasant"*.

### C.3.1 RESULTS

Table 14: CSI Scores for Word Pairs Across Models

| Model | Word Pair | O_score | P_score | N_score | Consist. R | Reluct. R |
|---|---|---|---|---|---|---|
| GPT-4o | Comedy/Tragedy | 0.4792 | 0.2726 | 0.2482 | 0.7536 | 0.0400 |
| | Good/Bad | 0.4342 | 0.0892 | 0.4766 | 0.7984 | 0.3747 |
| | Enjoyable/Unpleasant | 0.4442 | 0.1968 | 0.3590 | 0.7262 | 0.2010 |
| Qwen2-72B | Comedy/Tragedy | 0.5964 | 0.2314 | 0.1722 | 0.8280 | 0.0028 |
| | Good/Bad | 0.6430 | 0.1522 | 0.2048 | 0.8104 | 0.0872 |
| | Enjoyable/Unpleasant | 0.5462 | 0.3056 | 0.1482 | 0.8526 | 0.0180 |
| Llama3.1-70B | Comedy/Tragedy | 0.4492 | 0.3056 | 0.2452 | 0.7552 | 0.0055 |
| | Good/Bad | 0.7410 | 0.1760 | 0.0830 | 0.9180 | 0.0074 |
| | Enjoyable/Unpleasant | 0.5410 | 0.3144 | 0.1446 | 0.8568 | 0.0093 |

### C.3.2 OBSERVATIONS

Using strongly negative words like *bad"* (compared to *tragedy"*) triggered the models' guardrails, causing them to avoid negative associations. For instance, GPT-4o's Pessimism score dropped significantly from 0.2726 to 0.0892 with *bad"*, while Neutrality increased from 0.2482 to 0.4766. In contrast, milder terms like *unpleasant"* had less impact on scores, demonstrating CSI's robustness when following our word selection principles.

Across all settings, CSI maintained strong reliability metrics (**Consistency** and **Reluctant**), consistently outperforming traditional BFI scores. The only exception was GPT-4o showing a higher Reluctant rate with the *good" / bad"* pair, further supporting our principle of avoiding strongly triggering terms.

Table 15: BFI Scores Comparison (Consistency and Reluctant)

| Model | Consistency | Reluctant |
|---|---|---|
| GPT-4o | 0.5227 | 0.1477 |
| Qwen2-72B-Instruct | 0.6818 | 0.0909 |
| Llama 3.1-70B-Instruct | 0.5227 | 0.0568 |

These results confirm that while word choice can influence CSI scores, adhering to our word selection principles yields robust and reliable results across models and settings, consistently outperforming traditional BFI measurements.

## C.4 CROSS-LINGUAL EVALUATIONS

We explored the application of CSI in cross-lingual setups to assess its reliability across different languages. Experiments were conducted using the Qwen2-72B-Instruct model.

### C.4.1 RESULTS

Table 16: Monolingual CSI Scores for Qwen2-72B-Instruct

| Language | O_score | P_score | N_score | Consist. R | Reluct. R |
|----------|---------|---------|---------|------------|-----------|
| English  | 0.5964  | 0.2314  | 0.1722  | 0.8280     | 0.0028    |
| Chinese  | 0.5312  | 0.2736  | 0.1952  | 0.8050     | 0.0134    |

**Monolingual Evaluations with Qwen2-72B-Instruct**

Table 17: Cross-Lingual CSI Scores for Qwen2-72B-Instruct

| Prompt/Response | O_score | P_score | N_score | Consist. R | Reluct. R |
|-----------------|---------|---------|---------|------------|-----------|
| Chinese / English | 0.5216 | 0.2778 | 0.2006 | 0.7994 | 0.0035 |
| English / Chinese | 0.4992 | 0.3114 | 0.1894 | 0.8106 | 0.0036 |

**Cross-Lingual Prompting Scenarios**

### C.4.2 OBSERVATIONS

The model's scores in cross-lingual setups are comparable to those in monolingual evaluations, with no significant differences observed. Both **Consistency** and **Reluctant** rates remain excellent across all scenarios, indicating that CSI maintains high reliability even when prompts and responses are in different languages.

These findings demonstrate that CSI is effective and reliable in cross-lingual contexts, further validating its applicability for evaluating multilingual language models.

## C.5 SUMMARY

In summary, CSI delivers consistent results under varying parameters, including the number of items ($N$), temperature settings, and word pair selections. Additionally, CSI's reliability metrics (**Consistency** and **Reluctant**) consistently outperform traditional BFI methods across all tested configurations. These results confirm that CSI is a robust tool for evaluating language models, offering reliable measurements even in cross-lingual contexts.

# D MODEL DIAGNOSIS REPORT

## D.1 NUMERICAL REPORTS

## D.2 QUALITATIVE REPORTS

| Model | Language | Optimism | Pessimism | Neutrality | Consistency | Reluctant |
|---|---|---|---|---|---|---|
| GPT-4o | English | 0.4792 | 0.2726 | 0.2482 | 0.7536 | 0.0400 |
| GPT-4o | Chinese | 0.4786 | 0.2470 | 0.2744 | 0.7282 | 0.0483 |
| GPT-4 (1106) | English | 0.4658 | 0.2642 | 0.2700 | 0.7408 | 0.0871 |
| GPT-4 (1106) | Chinese | 0.6524 | 0.1934 | 0.1542 | 0.8462 | 0.0125 |
| GPT-4 (0125) | English | 0.5732 | 0.2638 | 0.1630 | 0.8370 | 0.0025 |
| GPT-4 (0125) | Chinese | 0.6256 | 0.2098 | 0.1646 | 0.8358 | 0.0033 |
| GPT-3.5 Turbo | English | 0.7328 | 0.1288 | 0.1384 | 0.8616 | 0.0000 |
| GPT-3.5 Turbo | Chinese | 0.6754 | 0.1598 | 0.1648 | 0.8352 | 0.0038 |
| Qwen2-72B | English | 0.5964 | 0.2314 | 0.1722 | 0.8280 | 0.0028 |
| Qwen2-72B | Chinese | 0.5312 | 0.2736 | 0.1952 | 0.8050 | 0.0134 |
| LLaMA 3.1 | English | 0.4492 | 0.3056 | 0.2452 | 0.7552 | 0.0055 |
| LLaMA 3.1 | Chinese | 0.2790 | 0.4794 | 0.2416 | 0.7584 | 0.0022 |

Table 18: Sentiment Scores and Reliability Metrics for all models.

| Model & Language | Top 20 Comedy Words | Top 20 Tragedy Words | Top 20 Neutral Words |
|---|---|---|---|
| gpt-3.5-turbo Chinese | 是, 可以, 我, 你, 我们, 有, 您, 会, 使用, 进行, 人, 为, 智能, 自己, 它, 提供, 技术, 能, 这, 发展 | 需要, 可能, 身体, 医疗, 世界, 要求, 导致, 控制, 情感, 历史, 风险, 能源, 污染, 感受, 价值, 压力, 生命, 必须, 疾病, 气候 | 问题, 让, 要, 数据, 文章, 影响, 其, 时间, 分析, 人类, 出, 情况, 社会, 考虑, 减少, 需求, 注意, 质量, 她, 没有 |
| gpt-3.5-turbo English | is, you, I, it, be, they, It, help, have, we, them, use, me, provide, he, she, information, make, using, used | impact, life, process, environment, challenges, issues, management, government, effects, end, security, risk, importance, safety, yourself, conditions, climate, prevent, times, healthcare | was, has, time, had, been, were, world, health, ensure, being, him, water, see, change, power, need, needs, know, areas, feel |
| gpt-4o Chinese | 是, 可以, 你, 我们, 有, 使用, 进行, 让, 它, 能, 这, 他们, 学习, 帮助, 他, 包括, 能够, 提高, 方式 | 需要, 会, 问题, 自己, 公司, 影响, 时间, 工作, 情况, 考虑, 减少, 身体, 没有, 医疗, 去, 世界, 要求, 导致, 结果, 任务 | 我, 您, 人, 为, 智能, 提供, 技术, 要, 数据, 发展, 到, 请, 选择, 环境, 信息, 文章, 其, 应用, 应该, 领域 |
| gpt-4o English | is, you, has, they, help, we, me, she, make, using, s, You, create, including, support, health, language, energy, example, ensure | was, them, time, had, provide, been, information, were, used, work, impact, world, media, being, system, reduce, research, change, power, environment | I, it, be, It, have, use, he, data, people, way, They, life, AI, him, water, process, development, practices, Use, her |
| gpt4-0125-preview Chinese | 是, 可以, 我, 你, 我们, 有, 您, 会, 使用, 进行, 人, 为, 智能, 自己, 让, 它, 提供, 技术, 能, 要 | 需要, 问题, 数据, 公司, 影响, 时间, 人类, 社会, 减少, 计算, 关系, 没有, 医疗, 世界, 要求, 导致, 结果, 存在, 控制, 函数 | 选择, 文章, 方式, 工作, 领域, 系统, 分析, 情况, 处理, 保护, 考虑, 以下, 研究, 需求, 代码, 注意, 她, 城市, 去, 其中 |
| gpt4-0125-preview English | is, you, I, it, be, has, they, help, have, we, them, use, me, provide, he, she, make, using, data, s | time, had, were, used, impact, world, health, life, being, system, research, power, industry, environment, challenges, body, issues, need, needs, years | was, It, been, information, ensure, examples, water, individuals, process, development, reduce, practices, change, resources, Use, add, based, others, story, code |
| gpt4-1106-preview Chinese | 是, 可以, 我, 你, 我们, 有, 您, 会, 使用, 进行, 人, 智能, 自己, 让, 它, 提供, 技术, 能, 要, 这 | 需要, 问题, 时间, 情况, 管理, 减少, 关系, 没有, 医疗, 要求, 导致, 结果, 函数, 避免, 情感, 利用, 历史, 风险, 投资, 经济 | 为, 到, 请, 公司, 他, 文章, 其, 应该, 领域, 系统, 想, 人类, 处理, 过程, 保护, 考虑, 确保, 需求, 计算, 成为 |
| gpt4-1106-preview English | you, it, be, It, help, we, them, use, he, she, make, s, people, You, way, create, including, They, life, language | I, time, had, used, data, impact, example, system, reduce, power, resources, environment, challenges, issues, others, code, need, needs, years, lead | is, was, has, they, have, me, provide, been, information, were, using, work, world, support, health, ensure, examples, water, She, individuals |
| llama3.1-70b-instruct Chinese | 我们, 有, 您, 会, 智能, 让, 能, 请, 帮助, 能够, 提高, 产品, 想, 可, 活动, 实现, 服务, 游戏, 对话, 健康 | 我, 需要, 使用, 问题, 进行, 人, 为, 它, 提供, 技术, 要, 这, 数据, 他们, 公司, 环境, 他, 信息, 文章, 影响 | 是, 可以, 你, 自己, 发展, 到, 学习, 选择, 包括, 建议, 应该, 可能, 设计, 人类, 处理, 能力, 保持, 确保, 语言, 写 |
| llama3.1-70b-instruct English | is, you, I, it, be, has, they, It, help, we, me, provide, he, she, make, people, way, create, They, support | time, had, been, were, impact, ensure, AI, him, individuals, system, process, reduce, research, change, power, industry, environment, challenges, body, issues | was, have, them, use, information, using, used, data, s, You, work, including, world, health, life, media, example, examples, experience, made |
| qwen2-72b-instruct Chinese | 是, 可以, 我, 你, 我们, 有, 您, 会, 使用, 人, 为, 智能, 自己, 让, 提供, 能, 要, 这, 发展, 他们 | 需要, 问题, 数据, 环境, 时间, 工作, 领域, 分析, 文化, 考虑, 管理, 减少, 研究, 需求, 质量, 没有, 医疗, 要求, 导致, 结果 | 进行, 它, 技术, 公司, 他, 影响, 方法, 方面, 应该, 系统, 用户, 人类, 情况, 社会, 过程, 保护, 确保, 写, 代码, 计算 |
| qwen2-72b-instruct English | is, you, I, it, be, was, has, It, help, have, we, use, had, me, he, she, information, make, were, using | time, work, impact, world, health, life, system, power, challenges, issues, need, needs, years, lead, business, changes, history, focus, control, government | they, them, provide, been, data, media, ensure, being, experience, technology, process, research, change, resources, industry, environment, body, areas, family, understanding |

Table 19: Top 20 Comedy, Tragedy, and Neutral Words Of Each Model.