# OpenReview forum: "Leveraging Implicit Sentiments: Enhancing Reliability and Validity in Psychological Trait Evaluation of LLMs"
_ICLR.cc/2025/Conference — Submitted to ICLR 2025_

### Official Review · Reviewer_QMAg · 2024-10-29

**Soundness:** 3
**Presentation:** 2
**Contribution:** 2
**Rating:** 5
**Confidence:** 4

**Summary:**

The paper introduces the Core Sentiment Inventory (CSI), a new evaluation method inspired by the Implicit Association Test (IAT) to assess the implicit sentiment tendencies of large language models (LLMs). The approach aims to provide a reliable and valid measure of LLMs' optimism, pessimism, and neutrality in both English and Chinese, surpassing conventional human-centric psychometric tests like the Big Five Inventory (BFI). The authors present experimental results that claim improved reliability, reduced reluctance rates, and strong predictive power for CSI.

**Strengths:**

CSI represents an interesting attempt to create a psychometric assessment tool tailored for LLMs, addressing concerns around LLM reluctance and consistency with human-designed psychometric scales.

The bilingual approach (English and Chinese) is a notable effort to capture linguistic and cultural variance in model behaviors, which is increasingly important as LLMs are deployed globally.

The experiments cover several dimensions of reliability and validity, with additional sentiment analyses through story generation, providing a range of quantitative metrics.

**Weaknesses:**

The paper does not sufficiently justify the underlying premise that implicit sentiment tests designed for humans (like IAT) can meaningfully assess non-human entities like LLMs. The model’s association of specific words with positive or negative sentiments may not translate into meaningful or actionable insights about its “psychological traits,” as LLMs lack actual consciousness or subjective experience.

CSI is evaluated solely through internal metrics without external validation from human experts in psychometrics or linguistics. Given the novelty of the tool, expert evaluation is essential to substantiate the claims of reliability and practical value, particularly for a method positioned as a "psychological trait" assessment.

The word set of 5,000 items lacks diversity and cultural depth, and it is unclear how these words were chosen or if they were screened for cultural or contextual biases. This oversight introduces potential biases that could skew CSI’s predictive power and undermine its reliability across varied contexts.

Many new mental health-based LLMs are not cited to show the differences  anf effectiveness of this paper.

**Questions:**

See above

---

> ### Author Response · Authors · 2024-11-21
> **Clarifying the Applicability and Methodology of CSI (1/2)**
>
> **comment 1:**
>
> *The paper does not justify that implicit sentiment tests designed for humans can meaningfully assess LLMs, which lack consciousness.*
>
> **Response:**
>
> While LLMs do not possess consciousness, they are trained on extensive amounts of human-generated text, enabling them to model complex patterns and associations present in human language, including those related to sentiment and psychological traits [1]. This training allows LLMs to exhibit behaviors that are functionally analogous to human responses in certain contexts.
>
> To justify the applicability of implicit sentiment tests to LLMs, we conducted an external validation task detailed in Section 4.3 (**RQ3: Validity Assessment**). In this task, the models were prompted to generate stories based on words from the CSI. We found a strong correlation between the sentiment of the generated stories and the proportion of negative words used as prompts (figure 3). This result demonstrates that CSI assessments translate into observable model behaviors, indicating that implicit sentiment tests can meaningfully assess LLMs despite their lack of consciousness.
>
> [1] Max Pellert, Clemens M Lechner, Claudia Wagner, Beatrice Rammstedt, and Markus Strohmaier. AI psychometrics: Using psychometric inventories to obtain psychological profiles of large language models. PsyArXiv, 2023. doi: 10.31234/osf.io/jv5dt URL https://doi.org/10.31234/osf.io/jv5dt
>
> **comment 2:**
>
> *CSI is evaluated solely through internal metrics without external validation from human experts in psychometrics or linguistics.*
>
> **Response:**
>
> We agree that external validation from human experts in psychometrics and linguistics could enhance the robustness of our study. However, traditional psychometric evaluation methods are primarily designed for assessing conscious human subjects and may not fully capture the unique characteristics of AI systems like LLMs. Our implementation of human-designed scales such as the BFI revealed significant limitations, including high reluctance rates and inconsistent responses.
>
> Our work aims to pioneer AI-specific assessment methodologies tailored to the distinctive nature of LLMs. By developing the CSI, we are taking an initial step toward creating evaluation tools that account for the differences between human cognition and AI language modeling. We believe that innovative methods like CSI can provide valuable insights into AI behavior, generating unique perspectives that traditional human-centric assessment tools might not offer.
>
> We acknowledge that future research would greatly benefit from interdisciplinary collaboration between AI researchers and experts in psychology and linguistics. Such partnerships could lead to the development of more refined assessment methodologies, ensuring that AI evaluation tools are both scientifically rigorous and contextually appropriate. We are committed to pursuing such collaborative efforts in our future work.
>
>
>
> **comment 3:**
>
>
>
> *The word set of 5,000 items lacks diversity and cultural depth, and it’s unclear how these words were chosen.*
>
> **Response:**
>
> Thank you for pointing this out. We assure you that our word selection process was both rigorous and systematic, designed to maximize diversity and cultural depth. As detailed in Section 3.3, we adhered to two key principles:
>
> ​	1.	**Neutral Word Selection:** We selected nouns and verbs to avoid inherent sentiment, minimizing emotional bias and ensuring a more balanced and objective assessment.
>
> ​	2.	**Data-Driven Approach for Representativeness:** Our word set was curated from large, real-world corpora and datasets, reflecting authentic language usage across various contexts, cultures, and domains. This includes sources such as UltraChat, Alpaca-GPT4, WizardLM-Evol-Instruct-V2-196K, llm-sys, and BELLE-MultiTurn-Chat and so on, along with over 16 other large-scale popular datasets. These sources encompass a wide range of linguistic expressions, enhancing the cultural depth and diversity of our word set.
>
> By employing this data-driven methodology and expanding the word set to **5,000 items**, we significantly increased linguistic coverage compared to traditional psychometric scales, which typically contain fewer than **100 items** (as shown in Table 1). This extensive coverage helps to capture a more comprehensive representation of language, thereby minimizing cultural and contextual biases that might arise from subjective word selection.

---

> ### Author Response · Authors · 2024-11-21
> **Clarifying the Applicability and Methodology of CSI (2/2)**
>
> **comment 4:**
>
> *Many new mental health-based LLMs are not cited.*
>
> **Response:**
>
> Thank you for bringing this to our attention. Large language models in mental health show promise for education, assessment, and intervention, highlighting the critical need for model behavior evaluation. This aligns closely with our CSI. We will include relevant references to recent work in mental health applications of LLMs to strengthen the context of our paper and you are welcomed to provide more paper, we'd happy to discuss in our paper.
>
> Stade, Elizabeth C., et al. "Large language models could change the future of behavioral healthcare: a proposal for responsible development and evaluation." *NPJ Mental Health Research* 3.1 (2024): 12.
>
> Lawrence, Hannah R., et al. "The opportunities and risks of large language models in mental health." *JMIR Mental Health* 11.1 (2024): e59479.
>
> Guo, Zhijun, et al. "Large Language Models for Mental Health Applications: Systematic Review." *JMIR mental health* 11.1 (2024): e57400.

---

### Official Review · Reviewer_w2P7 · 2024-11-03

**Soundness:** 3
**Presentation:** 3
**Contribution:** 2
**Rating:** 6
**Confidence:** 3

**Summary:**

The authors propose Core Sentiment Inventory, a psychometric evaluation framework to assess LLMs' sentiment tendencies.

The experimental setup covers two languages, English and Chinese, 2 open-weight LLMs (LLama3.1-70B and Qwen2-72B), and 4 closed/proprietary LLMs (GPT-4o, two GPT4 checkpoints, and GPT-3.5).

The CSI consists of the 5k most frequent emotionally neutral words; it is assumed that noun and verbs are neutral, and thus the CSI word lists consist of the top-5k noun/verbs in the corpora.

The LLM is then provided with N words picked from the wordlist, and asked to to associate each word with "comedy" or "tragedy", thus revealing a sentiment bias for each word.

**Strengths:**

The paper proposes an elegant approach based on the adaptation of existing psychometric tools (IAT).

The paper is well written and easy to follow.

**Weaknesses:**

Although an enjoyable read, the work falls short when it comes to the experimental setup.

First, while I consider the proposed method more elegant and effective than human-tailored alternatives such as BFI, I am not convinced by the preliminary reliability tests conducted: it can be argued than reluctance is due to post-training strategies (e.g. guardrails, instruction-tuning), thus a different choice of (accessible) LLMs could have been more convincing -- e.g. the first Mistral release.

Second, some design choices seem discretional and not thoroughly justified: for instance, the choice of the words "comedy" / "tragedy" used as classes seems arbitrary.

**Questions:**

- did you consider using alternatives to "comedy" / "tragedy" (e.g. "fun", "good", "dramatic", "bad")?

- did you consider LLMs with no significant guardrails (such as, if I remember correctly, the first Mistral)?

- all models used in the paper are multilingual, have you considered prompting in cross-lingual setups (e.g. chinese prompt for english CSI) for additional reliability indications?

---

> ### Author Response · Authors · 2024-11-21
> **Testing Robustness and Reliability in CSI Through Strategic Word Selection and Cross-Lingual Validation (1/2)**
>
> We appreciate the insightful feedback provided by the reviewers. Below, we address each point raised.
>
> **Comment 1:**
>
> *Reluctance may be due to post-training strategies like guardrails; using LLMs without such features could be more convincing.*
>
> **Response:**
>
> Thank you for highlighting this important consideration. We acknowledge that understanding the precise origins of reluctance is a valuable research direction. However, our primary contribution lies in demonstrating that **CSI significantly reduces reluctance rates compared to traditional personality scales like the BFI**, achieving near-zero reluctance rates versus up to 80% with conventional methods.
>
> Following the suggestion, we tested Mistral-7B-v0.1 on CSI. To note that, Mistral-7B-v0.1 is a non-chat model which can only be used as text completion, so we adjusted the prompts accordingly. However, due to its limited instruction-following capabilities, the model often echoed the provided test words without generating valid responses, making it difficult to analyze results.
>
> We believe that comparing identical models with and without guardrails would provide clearer insights into the influence of guardrails on reluctance. Since our current tests showed no significant differences in CSI scores across other models, we propose to explore this investigation in future work when more suitable resources become available.
>
>
>
>
>
> **Comment 2:**
>
> *The choice of “comedy” / “tragedy” seems arbitrary.*
>
> **Response:**
>
> Our selection of the word pair **“comedy” / “tragedy”** was guided by two key principles:
>
> ​	1.	**Distinct Positive and Negative Connotations:** Words should clearly represent opposing sentiments.
>
> ​	2.	**Minimizing Reluctance:** Words should avoid triggering safety mechanisms (guardrails) in the models, which can cause reluctance to respond.
>
> To address your concern, we conducted an ablation study using alternative word pairs: **Comedy / Tragedy**, **Good / Bad**, and **Enjoyable / Unpleasant**.
>
> In the word pair **“Good” / “Bad”**, “bad” conveys negativity more directly than “tragedy.” In contrast, in the word pair **“Enjoyable” / “Unpleasant”**, “unpleasant” conveys negativity less directly than “bad.”
>
> We compared these word pairs to examine their impact on CSI scores and reliability metrics.
>
> ### CSI Scores **Across** Different Word pairs:
>
> | **Model**                  | **Word Pair**          | **Optimism** | **Pessimism** | **Neutrality** | **Consistency** | **Reluctance** |
> | -------------------------- | ---------------------- | ------------ | ------------- | -------------- | --------------- | -------------- |
> | **GPT-4o**                 | Comedy / Tragedy       | 0.4792       | 0.2726        | 0.2482         | 0.7536          | 0.0400         |
> |                            | Good / Bad             | 0.4342       | 0.0892        | 0.4766         | 0.7984          | 0.3747         |
> |                            | Enjoyable / Unpleasant | 0.4442       | 0.1968        | 0.3590         | 0.7262          | 0.2010         |
> | **Qwen2-72B-Instruct**     | Comedy / Tragedy       | 0.5964       | 0.2314        | 0.1722         | 0.8280          | 0.0028         |
> |                            | Good / Bad             | 0.6430       | 0.1522        | 0.2048         | 0.8104          | 0.0872         |
> |                            | Enjoyable / Unpleasant | 0.5462       | 0.3056        | 0.1482         | 0.8526          | 0.0180         |
> | **Llama 3.1-70B-Instruct** | Comedy / Tragedy       | 0.4492       | 0.3056        | 0.2452         | 0.7552          | 0.0055         |
> |                            | Good / Bad             | 0.7410       | 0.1760        | 0.0830         | 0.9180          | 0.0074         |
> |                            | Enjoyable / Unpleasant | 0.5410       | 0.3144        | 0.1446         | 0.8568          | 0.0093         |
>
>
>
> ### BFI Scores Comparison (Consistency | Reluctant)
>
>
> | Model                 | Consistency | Reluctant |
> | --------------------- | ----------- | --------- |
> | GPT-4o                | 0.5227      | 0.1477    |
> | Qwen2-72B-instruct    | 0.6818      | 0.0909    |
> | Llama3.1-70B-instruct | 0.5227      | 0.0568    |

---

> ### Author Response · Authors · 2024-11-21
> **Testing Robustness and Reliability in CSI Through Strategic Word Selection and Cross-Lingual Validation (2/2)**
>
> **Observations**:
>
> First, using strongly negative words like "bad" (compared to "tragedy") triggered models' guardrails, causing them to avoid negative associations. For instance, GPT-4o's pessimism score dropped significantly from 0.2726 to 0.0892 with "bad", while neutrality increased from 0.2482 to 0.4766. In contrast, milder terms like “unpleasant” had less impact on scores, demonstrating CSI’s robustness when following our word selection principles.
>
> Second, across all settings, CSI maintained strong reliability metrics (**Consistency** and **Reluctance**), consistently outperforming traditional BFI scores. The only exception was GPT-4o showing a higher reluctance rate with the Good/Bad pair, further supporting our principle of avoiding strongly triggering terms.
>
> These results confirm that while word choice can influence CSI scores, adhering to our word selection principles yields robust and reliable results across models and settings, consistently outperforming traditional BFI measurements.
>
>
>
>
>
> **comment 3:**
>
> *Have you considered prompting in cross-lingual setups for additional reliability indications?*
>
> **Response:**
>
> Thank you for your insightful suggestion regarding cross-lingual setups for additional reliability indications. We found this idea intriguing and conducted experiments using the Qwen2-72B-Instruct model to assess CSI’s reliability in cross-lingual contexts.
>
> **Results:**
>
> 1. Monolingual Evaluations with Qwen2-72B-Instruct
>
> | Evaluation Language | Comedy Score | Pessimism | Neutrality | Consistency | Reluctance |
> | ------------------- | ------------ | --------- | ---------- | ----------- | ---------- |
> | English             | 0.5964       | 0.2314    | 0.1722     | 0.8280      | 0.0028     |
> | Chinese             | 0.5312       | 0.2736    | 0.1952     | 0.8050      | 0.0134     |
>
> 2. Cross-Lingual Prompting Scenarios
>
> | Prompt Language | Response Language | Comedy Score | Tragedy Score | Neutrality Score | Consistency Rate | Reluctance Rate |
> | --------------- | ----------------- | ------------ | ------------- | ---------------- | ---------------- | --------------- |
> | Chinese         | English           | 0.5216       | 0.2778        | 0.2006           | 0.7994           | 0.0035          |
> | English         | Chinese           | 0.4992       | 0.3114        | 0.1894           | 0.8106           | 0.0036          |
>
> The model’s scores in cross-lingual setups are comparable to those in monolingual evaluations, with no significant differences observed. Both **Consistency** and **Reluctance** rates remain excellent across all scenarios, indicating that CSI maintains high reliability even when prompts and responses are in different languages. These findings demonstrate that CSI is effective and reliable in cross-lingual contexts, further validating its applicability for evaluating multilingual LLMs.

---

### Official Review · Reviewer_4sSm · 2024-11-04

**Soundness:** 2
**Presentation:** 2
**Contribution:** 2
**Rating:** 3
**Confidence:** 4

**Summary:**

This paper introduces a novel assessment method known as the Core Sentiment Inventory (CSI) for evaluating the emotional tendencies of large language models (LLMs). Inspired by the Implicit Association Test (IAT), the CSI aims to provide a more reliable and effective way to assess the implicit emotional characteristics of LLMs. It addresses the limitations of traditional psychometric methods, such as model reluctance and inconsistency.

**Strengths:**

•  The CSI can effectively quantify the emotions of models, reflecting the emotional tendencies of LLMs.

•  It effectively reduces the issues of reluctance and inconsistency in traditional psychological scales.

•  The use of representative neutral words in constructing the CSI reduces the potential emotional orientation of the words themselves, better reflecting the internal emotional associations of LLMs.

•  It explores the impact of different language environments on the emotional tendencies of LLMs.

**Weaknesses:**

•  The constructed CSI is only used to assess the emotions of LLMs and has not been extended to other psychometric fields, such as personality or moral decision-making.

•  It does not introduce the calculation methods for consistency rate and reluctance rate.

**Questions:**

•  Discuss how CSI can improve existing psychological scales designed for humans.

•  Further explore the unintentional biases that language models may have towards everyday concepts as mentioned in section 4.1.

•  On line 424, it might be: “e.g., five positive words, four neutral words and one negative words, and so on.”

---

> ### Author Response · Authors · 2024-11-20
> **Clarifying CSI as an Alternative Evaluation Tool to Existing Methods**
>
> We appreciate the review. Below, we address each point raised.
>
>
>
>
>
> **Comment 1: Limited Extension to Other Psychometric Fields**
>
> *The CSI is only used to assess emotions of LLMs and hasn’t been extended to fields like personality or moral decision-making.*
>
>
>
> **Response:**
>
> We acknowledge that CSI currently focuses on sentiment assessment. Our work serves as a foundational step in developing psychometric tools specifically for LLMs. Extending this approach to other domains like personality or moral decision-making is a promising direction for future research, which may require developing additional specialized tools akin to CSI.
>
>
>
> **Comment 2:**
>
> *The paper does not introduce the calculation methods for consistency rate and reluctance rate.*
>
>
>
> **Response:**
>
> We apologize for any confusion. In Section 4.2, we describe:
>
> *"The consistency rate measures the proportion of items where the model’s responses remained consistent across repeated trials. A higher consistency rate indicates greater reliability. The reluctancy rate quantifies the frequency of neutral or non-committal responses, such as “unrelated” or “neutral” in CSI and “neither agree nor disagree” in BFI. Higher reluctance indicates lower reliability."*
>
> In short:  Consistency Rate = (Number of Consistent Items) / (Total CSI Items (5,000)); Reluctance Rate = (Number of Reluctant Responses) / (Total CSI Items (5,000))
>
>
>
> **Question 1:**
>
> *Discuss how CSI can improve existing psychological scales designed for humans.*
>
> **Response:**
>
> To clarify, CSI is a tool developed specifically for evaluating LLMs, built independently of traditional psychological scales designed for humans. Unlike human-focused scales, CSI addresses challenges unique to AI systems, such as reluctance and consistency, making it a parallel approach rather than an extension of existing methodologies. As a framework built from ground up, CSI is not intended to improve human psychological scales but instead offers an alternative approach tailored to the unique requirements of AI evaluation.
>
>
>
> **Question 2:**
>
> *Further explore the unintentional biases that language models may have towards everyday concepts.*
>
> **Response:**
>
> We appreciate this suggestion. From educational guesses, unintentional biases in LLMs often arise from the extensive and diverse datasets they are trained on, which can inadvertently reflect societal biases embedded in the text. Addressing these biases is critical for building fair and equitable AI systems. Future research could focus on systematically analyzing the sources of these biases and developing robust mitigation strategies. While this is an important area of exploration, our current resources are limited. We aim to prioritize this direction in future work as additional resources become available.
>
>
>
> **Question 3:**
>
> *On line 424, it might be: “e.g., five positive words, four neutral words and one negative word, and so on.”*
>
> **Response:**
>
> We apologize for any confusion. To clarify, we selected five words as seeds for the model to generate stories. These seeds consist only of positive and negative words, without including neutral words.
>
> We consider rewriting the description as follows:
>
> "For example, all five positive words, four positive and one negative, three positive and two negative, and so on."

---

### Official Review · Reviewer_jSVf · 2024-11-05

**Soundness:** 2
**Presentation:** 4
**Contribution:** 2
**Rating:** 3
**Confidence:** 3

**Summary:**

The paper introduces Core Sentiment Inventory (CSI), a multilingual evaluation benchmark aimed at assessing the sentiment tendencies of LLMs in an implicit manner. The approach leverages 5,000 neutral words from English and Chinese and prompts the LLM to express polarity towards these neutral words. By assessing this polarity, the paper measures the biases of the models with respect to optimism or pessimism. To quantify the reliability of CSI, the paper validates the method against BFI, where CSI shows significant decrease in reluctance (i.e., model punting).

**Strengths:**

Reasons to accept
- The presentation of the method is clear and concise.
- Measuring LLM’s sentiment tendencies is vital to identify biases and building fair systems.
- The evaluation is performed both in English and Chinese with the approach being easily extended to other languages.

**Weaknesses:**

Reasons to reject

While the paper has merit, I see the some critical flaws presented below:

- The decision to pick all top nouns/verbs is questionable to me. Yes, nouns and verbs “tend” to be neutral. However, this is not always the case. From the examples in Table 2, some of these words are clearly polarized. “Improve” has positive connotations, as well as “team”. I believe there needs to be a manual filtering step where these words are removed to ensure reliable results. As it stands, I think the model does not have implicit biases if it assigns “improve” as positive.
- Design choices are not well-motivated. The approach does multiple predictions for the same word and the method shuffles the order of words to measure inconsistency. (1) Given this goal, why is temperature T set to 0? Wouldn’t a higher temperature better indicate the model uncertainty in assigning tragedy/comedy? (2) Why is the number of words sampled equal to 30? What happens with n > 30 or n < 30. Why wasn’t the number of words picked so it maximizes the context window?
- The prompt design is biased. (3) Why is neutral not a valid option? A very strong model with perfect instruction following capabilities will always pick one of the two (comedy/tragedy) and will never output “neutral”. (4) Given the definition of neutral score as N_{inconsistent} / N, I am wondering what percentage of words the model predicted in opposite categories? I think they should be very few. In this case, neutral score is solely determined by poor instruction following, not implicit biases.

**Questions:**

See numbered questions above: 1, 2, 3, 4

---

> ### Author Response · Authors · 2024-11-20
> **Clarifying CSI Construction Principles and Extended Ablation Studies (1/2)**
>
> We appreciate the insightful feedback provided by the reviewers. Below, we address each point raised.
>
>
>
> **Comment 1: Selection of Nouns and Verbs**
>
> *The decision to pick all top nouns/verbs is questionable. Some words like “improve” and “team” have positive connotations. There should be a manual filtering step to remove polarized words to ensure reliable results.*
>
> **Response:**
>
> Our selection of nouns and verbs is based on the principle that in natural language, sentiment is predominantly conveyed by modifiers (adjectives and adverbs) rather than heads (nouns and verbs) (Baccianella et al., 2010)[1]. This suggests that nouns and verbs are generally more neutral and suitable for our study.
>
> We acknowledge that some nouns and verbs may still carry emotional connotations. However, language is inherently subjective, and even seemingly positive words can carry negative associations depending on context which means that manually filtering these words could introduce human biases and undermine the objectivity of our tool.
> To avoid introducing human bias, we chose not to manually filter nouns and verbs. This approach allows CSI to reveal the models’ implicit associations naturally. For example, GPT-4o perceived words like “authority,” “nation,” and “oils” negatively, highlighting the value of our method in uncovering nuanced biases.
>
>
> **Comment 2:**
>
> *(1) Why is temperature T set to 0? Wouldn’t a higher temperature better indicate the model uncertainty in assigning tragedy/comedy? (2) Why is the number of words sampled equal to 30? What happens with n > 30 or n < 30?*
>
> **Response:**
>
> We used a temperature of 0, a standard setting in this field, to ensure reproducibility. A 30-word sampling was chosen for two main reasons: 1. **Token Consumption Control**: CSI has a total of 5000 words, and a bigger sampling number helps manage token usage efficiently. Since these models have different maximum context windows, 30 is a conservative choice, but increasing it could further improve cost-efficiency. 2. **Comparability**: This setup aligns with existing studies, such as PsychoBench [2], specifically in Section 4.2 RQ2: Reliability Assessment, allowing for direct comparison of results.
>
> Inspired by your review, we conducted ablation studies of CSI with **gpt-4o**, **llama3.1-70b-instruct**, and **qwen2-72b-instruct**, adjusting N and temperature respectively.
>
> First, we adjusting N while keeping the temperature fixed at 0:
>
> ### CSI Score with different item numbers within each input (Temperature = 0)
>
> #### GPT-4o
>
> | N    | Optimism | Pessimism | Neutrality | Consistency | Reluctant |
> | ---- | -------- | --------- | ---------- | ----------- | --------- |
> | 10   | 0.5048   | 0.3098    | 0.1854     | 0.8146      | 0.0010    |
> | 20   | 0.5292   | 0.2754    | 0.1954     | 0.8046      | 0.0017    |
> | 30   | 0.4792   | 0.2726    | 0.2482     | 0.7536      | 0.0400    |
> | 50   | 0.5540   | 0.2552    | 0.1908     | 0.8092      | 0.0045    |
> | 100  | 0.5486   | 0.2392    | 0.2122     | 0.7878      | 0.0001    |
>
> ---
>
> #### Llama3.1-70B-Instruct
>
> | N    | Optimism | Pessimism | Neutrality | Consistency | Reluctant |
> | ---- | -------- | --------- | ---------- | ----------- | --------- |
> | 10   | 0.4158   | 0.3578    | 0.2264     | 0.7736      | 0.0025    |
> | 20   | 0.4298   | 0.3284    | 0.2418     | 0.7582      | 0.0073    |
> | 30   | 0.4492   | 0.3056    | 0.2452     | 0.7552      | 0.0055    |
> | 50   | 0.4518   | 0.2908    | 0.2574     | 0.7428      | 0.0068    |
> | 100  | 0.4918   | 0.2450    | 0.2632     | 0.7368      | 0.0066    |
>
> ---
>
> #### Qwen2-72B-Instruct
>
> | N    | Optimism | Pessimism | Neutrality | Consistency | Reluctant |
> | ---- | -------- | --------- | ---------- | ----------- | --------- |
> | 10   | 0.5646   | 0.2546    | 0.1808     | 0.8194      | 0.0043    |
> | 20   | 0.5682   | 0.2578    | 0.1740     | 0.8260      | 0.0013    |
> | 30   | 0.5964   | 0.2314    | 0.1722     | 0.8280      | 0.0028    |
> | 50   | 0.6068   | 0.2278    | 0.1654     | 0.8346      | 0.0008    |
> | 100  | 0.6466   | 0.1900    | 0.1634     | 0.8366      | 0.0000    |
>
> Pessimism, Optimism, and Neutrality are the three dimensions of the CSI score, while Consistency and Reluctance are two metrics for reliability.
>
> From the above table, we can see the absolute numbers of CSI scores show minor variations, with 30 as the baseline. Specifically, the Optimism score of each models are: **gpt-4o**: 0.4792 ± 0.07, **llama3.1-70b-instruct**: 0.4492 ± 0.05, **qwen2-72b-instruct**: 0.5964 ± 0.05.
>
> Importantly, the **Consistency** and **Reluctant** metrics remained stable across all settings, outperforming traditional methods like BFI significantly.
>
>
>
> ### BFI Scores Comparison (Consistency | Reluctant)
>
> | Model                 | Consistency | Reluctant |
> | --------------------- | ----------- | --------- |
> | GPT-4o                | 0.5227      | 0.1477    |
> | Qwen2-72B-instruct    | 0.6818      | 0.0909    |
> | Llama3.1-70B-instruct | 0.5227      | 0.0568    |

---

> ### Author Response · Authors · 2024-11-20
> **Clarifying CSI Construction Principles and Extended Ablation Studies (2/2)**
>
> Second, we further explored the impact of varying temperatures (0–1) with N fixed at 30:
>
> ### gpt-4o (N = 30, Temperature 0.0 - 1.0)
>
> | Temperature | Optimism | Pessimism | Neutrality | Consistency | Reluctant |
> | ----------- | -------- | --------- | ---------- | ----------- | --------- |
> | 0.0         | 0.4792   | 0.2726    | 0.2482     | 0.7536      | 0.0400    |
> | 0.1         | 0.5748   | 0.2770    | 0.1482     | 0.8518      | 0.0000    |
> | 0.3         | 0.5640   | 0.2816    | 0.1544     | 0.8456      | 0.0015    |
> | 0.5         | 0.5574   | 0.2728    | 0.1698     | 0.8302      | 0.0000    |
> | 0.7         | 0.5370   | 0.2778    | 0.1852     | 0.8148      | 0.0017    |
> | 0.99        | 0.5202   | 0.2752    | 0.2046     | 0.7954      | 0.0001    |
> | 1.0         | 0.5198   | 0.2800    | 0.2002     | 0.7998      | 0.0004    |
>
> ---
>
> ### qwen2-72b-instruct (N = 30, Temperature 0.0 - 1.0)
>
> | Temperature | Optimism | Pessimism | Neutrality | Consistency | Reluctant |
> | ----------- | -------- | --------- | ---------- | ----------- | --------- |
> | 0.0         | 0.5964   | 0.2314    | 0.1722     | 0.8280      | 0.0028    |
> | 0.1         | 0.5992   | 0.2350    | 0.1658     | 0.8346      | 0.0039    |
> | 0.3         | 0.5804   | 0.2452    | 0.1744     | 0.8258      | 0.0041    |
> | 0.5         | 0.5890   | 0.2410    | 0.1700     | 0.8300      | 0.0029    |
> | 0.7         | 0.5726   | 0.2520    | 0.1754     | 0.8246      | 0.0033    |
> | 0.99        | 0.5672   | 0.2486    | 0.1842     | 0.8160      | 0.0068    |
> | 1.0         | 0.5810   | 0.2524    | 0.1666     | 0.8334      | 0.0037    |
>
> ---
>
> ### llama3.1-70b-instruct (N = 30, Temperature 0.0 - 1.0)
>
> | Temperature | Optimism | Pessimism | Neutrality | Consistency | Reluctant |
> | ----------- | -------- | --------- | ---------- | ----------- | --------- |
> | 0.0         | 0.4492   | 0.3056    | 0.2452     | 0.7552      | 0.0055    |
> | 0.1         | 0.4412   | 0.3178    | 0.2410     | 0.7590      | 0.0040    |
> | 0.3         | 0.4428   | 0.3094    | 0.2478     | 0.7522      | 0.0083    |
> | 0.5         | 0.4370   | 0.3082    | 0.2548     | 0.7456      | 0.0048    |
> | 0.7         | 0.4156   | 0.3194    | 0.2650     | 0.7350      | 0.0089    |
> | 0.99        | 0.4050   | 0.3196    | 0.2754     | 0.7250      | 0.0138    |
> | 1.0         | 0.3902   | 0.3366    | 0.2732     | 0.7270      | 0.0084    |
>
> The results also show minimal variation in model behavior when calculating CSI across different temperatures.
>
>
>
> In summary, CSI delivers consistent results under varying parameters, sampling size and model temperature. Additionally, CSI’s reliability consistently outperforms traditional BFI methods across all tested configurations.
>
>
>
> **Comment 3: Prompt Design Bias**
>
> *Why is neutral not a valid option? A strong model will always pick one of the two options and never output “neutral”.*
>
> **Response:**
>
> Our design aligns with the principles of the Implicit Association Test (IAT), which measures implicit biases by requiring a choice between two distinct categories. Including a neutral option could encourage non-committal responses, reducing the test’s effectiveness and differentiation capacity. By forcing a choice, we aim to reveal the model’s implicit associations more effectively.
>
>
>
> **Comment 4: Neutral Score and Instruction Following**
>
> *Given the definition of the neutral score as N_{inconsistent} / N, what percentage of words did the model predict in opposite categories? Are neutral scores solely due to poor instruction following?*
>
> **Response:**
>
> The neutral score do represents the proportion of inconsistent responses across multiple trials and we found that  12.88% - 47.94% of items had **consistent negative responses** (tragedy, P_score in table 3). This significant proportion indicates the effectiveness of CSI in revealing nagative bias of LLMs. Therefore, we believe that the neutral score is not solely due to poor instruction following but reflects the model’s uncertainty or ambivalence in its associations.
>
>
>
>
>
>
>
> [1]Stefano Baccianella, Andrea Esuli, and Fabrizio Sebastiani. Sentiwordnet 3.0: An enhanced lexical resource for sentiment analysis and opinion mining. In LREC. European Language Resources Association, 2010.
>
> [2]Jen-tse Huang, Wenxuan Wang, Eric John Li, Man Ho Lam, Shujie Ren, Youliang Yuan, Wenxiang Jiao, Zhaopeng Tu, and Michael R. Lyu. On the humanity of conversational AI: evaluating the psychological portrayal of llms. In ICLR. OpenReview.net, 2024.

---

### Author Response · Authors · 2024-11-29
**General Response**

We sincerely thank all the reviewers for their valuable feedback. Here, we further clarify our contributions:

​	1.	**Introduction of CSI as an AI-Specific Assessment Tool:** We propose the CSI as a behavior evaluation instrument that pioneers AI-specific assessment methodologies tailored to the distinctive nature of large language models (LLMs), providing an alternative to human-based psychometric scales.

​	2.	**Data-Driven and Objective Construction:** The construction of CSI is based on a data-driven approach with objective principles, making it a fair evaluation tool without introducing potential biases that might be caused by human subjective selection.

​	3.	**Broad Linguistic Coverage and Alignment with Model Usage:** This data-driven approach allows CSI to better align with model usage scenarios and covers a significantly broader linguistic range (5,000 items) compared to traditional psychometric scales, which typically have fewer than 100 items (e.g, BFI has 44 items).

​	4.	**Empirical Validation and Improved Reliability:** Our experimental results on several LLMs demonstrate that CSI successfully reveals nuanced patterns of LLM behavior in different scenarios and languages. Compared to the traditional BFI scale, CSI improves reliability by reducing reluctance rates and enhancing consistency rates. Furthermore, in validity tests using a story generation task, the results show that CSI assessments translate into observable model behaviors, indicating that CSI scores can meaningfully assess LLMs despite their lack of consciousness.



During the rebuttal period, we have:

​	1.	**Conducted Additional Ablation Studies:** We examined the impact of different sampling sizes (n) and temperature settings during testing to assess the robustness of CSI under varying conditions.

​	2.	**Explored the Effect of Word Selection:** We extended the original word pairs “comedy” / “tragedy” by including additional pairs such as “good” / “bad” and “enjoyable” / “unpleasant” to evaluate how word choice affects CSI scores and reliability metrics.

​	3.	**Evaluated Cross-Lingual Prompting Scenarios:** We assessed CSI scores in cross-lingual prompting scenarios, where prompts are provided in one language (English or Chinese) and the model’s responses are generated in the other language, to validate CSI’s applicability in multilingual contexts.

These additional experiments confirm that CSI delivers consistent results under varying parameters, including the number of items (N), temperature settings, and word pair selections. Additionally, CSI’s reliability metrics (Consistency and Reluctance) consistently outperform traditional BFI methods across all tested configurations. These results affirm that CSI is a robust tool for evaluating language models, offering reliable and meaningful measurements.

---

### Meta-Review · Area_Chair_do64 · 2024-12-21

**Metareview:**

**Summary:**

The authors introduce a bilingual evaluation (Chinese and English) for assessing LLM's implicit emotional characteristics. Their work builds on existing psychometric tests, and shows how LLMs vary in tendencies towards optimistic, pessimistic, and neutral behavior (there is a general bias towards optimism). They also show that language impacts the sentiment tendencies of LLMs, sometimes flipping their preferences from optimistic to pessimistic. Finally, they show that their approach is more reliable than a traditional psychometric test (BFI).

**Strengths:**

- In principal, this offers a simple and solid approach for quantifying how LLMs mimic human emotions.

- The multilingual setting allows for exploration of cultural variance in how sentiment is conveyed.

**Weaknesses:**

- The tone of the writing may over-anthrophomorize LLMs.

- The data was inadequately validated, for example the choice of words to measure sentiment seems less well-reasoned than would be desirable even after the rebuttal.

**Additional Comments On Reviewer Discussion:**

The reviews are mixed, but leaning towards rejection. From the rebuttal and discussion, there seem to be a number of unaddressed issues. Resolving these would significantly strengthen the work:

(1) Toning down the human-LLM psychology comparisons,

(2) Adding external validation from linguistics and psychometry experts,

(3) Adding more analysis of the word set like diversity.

I do commend the authors for exploring reviewer w2P7's suggestion of crosslingual prompting experiments, which will make the final version of the paper more comprehensive.

---

### Decision · Program_Chairs · 2025-01-22

Reject